# Emerging High-Depth Signal Tactile Sensing Technology for Achieving Artificial Human Touch

Edward Shim
Research & Engineering
Brighter Signals B.V.
Amsterdam, The Netherlands
edward@brightersignals.com

Mohammed Fazil Adam Basha
Engineering Department
University Concordia
Montreal, Canada
mohammedfazil.adambasha@
mail.concordia.ca

Hayden Choi
Computer Science
McMaster University
Hamilton, Canada
choih47@mcmaster.ca

Christine Fraser
Research & Engineering
Brighter Signals B.V.
Amsterdam, The Netherlands
christine@brightersignals.com

*Abstract* — **Over recent years, the state-of-the-art in tactile sensing has shifted towards visually measuring tactile membrane deformation. However, due to the multi-sensory nature of human touch, direct physical sensors remain beneficial. Traditionally, such sensors are often limited to singular measurements, have a limited signal range, and tend to have a binary (on-off) behavior.**

**The novel sensor described in this paper approaches this challenge through the substitution of non-recyclable polymers with a textile semi-conductive layer sandwiched between two conductive layers. This sensor provides a dynamic sensing range, even when the sensor deforms, similar to how humans feel a wide range of touch even when the fingers are curled or fully extended. With novel sensor-level filtering that reduces post-processing steps of data cleaning, signal filtering, and calibration, the result is continuous high-depth measurements for real-time identification of intended tactile stimuli, leading towards a breakthrough for achieving artificial human touch.**

*Keywords—tactile, textile, edge-computing, semi-conductor, sensing, multi-modal, robotics, automation, automotive, high-depth, emerging technology, soft robotics, fabric sensors, IOT*

## I. Introduction

Human touch could be understood as a multi-sensory experience that involves a combination of receptors detecting mechanical stimuli such as pressure, vibration, moisture, and temperature. To achieve this requires a skillful physical interaction of the hands [1] or other body surfaces, non-physical thermoreceptor responses such as feeling hot or cold [2], and changing environments such as how hands bend, fingertips curl, and fists clench.

Over recent years, there have been significant advancements in current state-of-the-art tactile sensing technologies, such as putting high-resolution cameras and ocular-based systems behind a tactile membrane [3]. However, these modalities remain inherently limited in their ability to replicate complex perceptual and interactive haptic feedback. Instead of solely relying on vision-based sensing that infers indirect interpretations of touch, sensors that actually measure physical stimuli in real-time deliver results that cannot be achieved by purely vision-based methods [4].

However, solutions that require separate and distinct components to directly measure force, pressure, temperature, and vibration introduce other challenges. Traditional sensors are often limited to single-purpose tasks due to their design, construction, and the structure of the materials used. They are limited in the range of measurements, requiring more and different sensors to detect additional parameters. The ability to replicate human-like touch in all its complexity with fewer parts, in a small form factor, would represent a quantum leap in endowing machines with a better understanding of the physical environment [5], thus enabling intelligent systems to improve their autonomous decision-making and execution capabilities [6].

## II. Applications of a New Sensor Material

Traditional tactile sensors use one of two fundamental principles for measuring the touch interaction. Capacitance-based sensors are made with a separating layer (insulator), an invisible wall for electrons. When the electrical field around the sensor changes, electrons can bridge that gap, allowing detection without physical contact. On the other hand, Resistive-based sensors operate on the principle of electrical resistance (conductor), which changes due to an external physical stimulus. These two distinct types of sensors have fundamentally different sensor constructions.

Our method introduces a semi-conductive, carbon-based textile layer that acts both as an insulator and a conductor, providing an alternative and improved tactile sensing capability. By reducing the number of sensors required, we achieve substantial benefits, including 1) Active-Dynamic sensing that combines vibration and pressure differential measurements, 2) multi-modal sensing with combined resistive and capacitance fundamentals, and 3) being able to conform to the physical environment without limitations of physical deformation that would otherwise result in sensor signal noise.

The high-depth sensitivity and the ability for the technology to detect minute signals corresponding to human physiology such as heart rate and respiration was proven in a published clinical trial [7].

## III. Active-Dynamic Versus Passive Sensing

In the context for this paper, the following terminology is used to describe sensing: Active-Dynamic, where a sensor is producing a continuous analog-like signal, and Passive sensing, where a sensor only produces a signal when there is a force applied producing a zero-state indicating "off" in the absence of force.

Existing examples of Passive sensors include Force Sensing Resistors (FSR) that measure force, and Bend/Flex (BF) sensors that measure angular displacement from physical deformation. These sensors have similar construction as they both utilize a polymer conductive film or pressure-sensitive layer with an air gap that bridges the

conductive layers by converting mechanical input into an electrical signal [8].

Physical stimuli bridge the air gap before it can sense an external force, hence the reason for the sensor being in an "off" state without external force. Figure 1 demonstrates Shunt Mode FSR, or conductive film in another type of Thru Mode FSR.

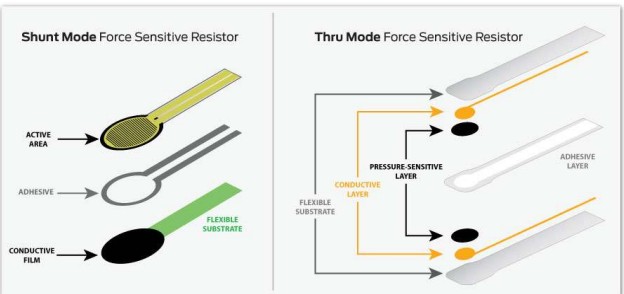

Fig 1: Example of how force sensing works using traditional technologies

We introduce a new method of Active-Dynamic sensing with a semi-conductive layer that provides higher electrical resistance when there is no external force, and lowering electrical resistance corresponding to increasing external force. Similar to the human sense of touch, the state of the sensor is continuously measuring high-depth signals just as how fingers actively feel without touching anything (low signal), and can feel an increasing range of applied force such as when squeezing an object or clenching a fist (low-to-high signal).

One of the well-known problems in producing FSR and BF sensors is that the polymer layer is crude, related to sensitivity, repeatability, and consistency, due to the inability to create conductive uniformity in the polymer of these sensors. This is due to the mixture of conductive particles used. This is especially true for polymers where non-uniformity has a strong influence on viscosity [9]; for them, non-homogeneous mixtures would be a critical factor influencing electrical and mechanical properties.

The semi-conductive layer as a direct substitute to the polymer layer has advantages where the semi-conductive layer is made of textiles that have a uniformly visible pattern of its conductive distribution as seen in Figure 2. This makes signal quality and signal error quantifiable. The fabrication of

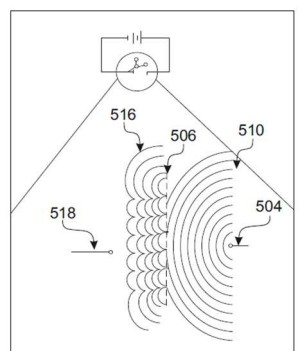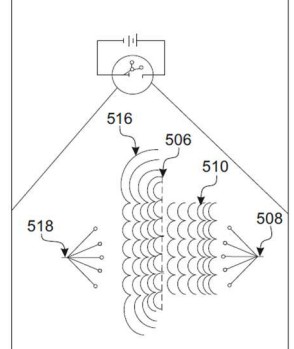

Fig 2: Example of uniform electrical dispersion 516 from output conductor 504 | 508, with electrical signal passing through the semi-conductor layer 506, where signal is proportional to physical stimuli of the input conductor.

the semi-conductive layer is described in Figures 5A and 5B of the referenced patent [10].

Due to the uniformity of the conductive behavior, the semi-conductive layer can measure external forces over a deep range, from a few grams to over a hundred kilograms, as a continuous signal requiring virtually no post-processing. This allows for several applications of the same sensor, reducing the need for a multitude of heterogeneous sensors.

## IV. MULTI-MODAL SENSING

The fabrication of the semi-conductive layer benefits with:
- The physical structure of the semi-conductive layer having visible uniformity for the conductive distribution,
- The semi-conductive layer acting as both an insulator and a conductor which allows multi-modal sensing with two fundamentally different modes of sensing in a single sensor through:
  a) Resistive measurements of Voltage differential ($\Delta V$) having a partially conductive layer, and
  b) Capacitive measurements having an insulating dielectric layer.

Sensing measurements using the resistive method allow measuring vibration (low amplitude change) and force (high amplitude change). This also provides increased sensitivity and higher depth signals with Active-Dynamic sensing.

Sensing measurements using the capacitive method enables the ability to detect proximity, contact of conductive objects, temperature using thermoelectric effect with different conductive materials, and/or humidity from capacitance change of the dielectric constant from water molecules [11].

## V. FUNCTIONING WITH CHANGING FORM FACTOR

Although flexible electronics have improved, limitations remain in the physical structure of the polymers. With both the FSR and BF sensors having similarities in their sensor construction, the BF sensor can measure force, an unintended feature, while a FSR can measure physical strain, also an unintended feature. This makes an argument that, in the presence of installation bending, the signal derived from an FSR and/or BF would at least have a 50% error rate in attributing the signal to the applied force or to the bending.

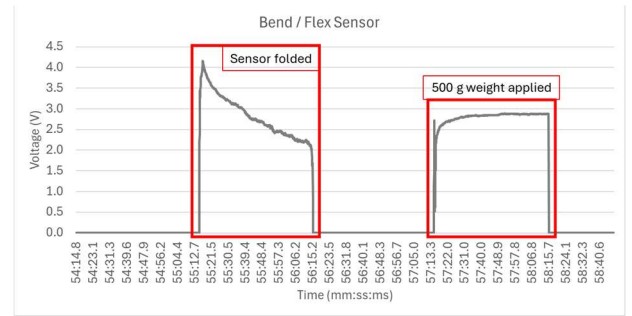

Fig 3: Bend / Flex sensor measuring parameters based on Voltage differential ($\Delta V$, $0 - 5$) at 60 second intervals per condition (Baseline -> Sensor Folded -> Baseline -> Weight Added -> Baseline).
The first; measuring bending in a curved teardrop shape with two ends touching each other, and the second; laid flat with 500 g weight applied.

Figure 3 demonstrates a BF sensor measuring both parameters of bending (Left) and force (Right). This shows a

significant limitation to maintain signal consistency when bending, with also the problem of differentiating when there is weight applied. This is not reliable for tactile sensing due to the inability to consistently measure a single mode, and the inability to differentiate between two states.

Figure 4 shows comparisons of a FSR and the new sensor with a textile semi-conductive layer in two conditions of 1) being laid out flat on a flat surface, and 2) being folded. The FSR was not assessed in the folded conditions due to the bending limitation, as demonstrated in Figure 3.

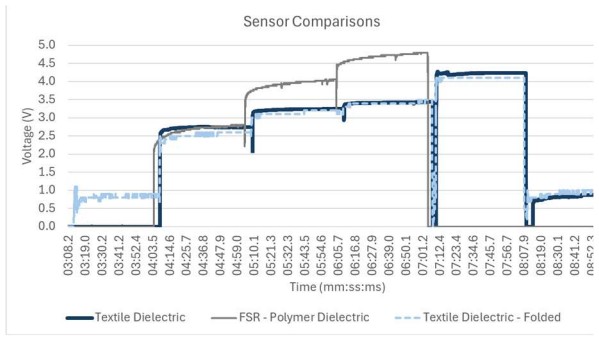

Fig 4: Tested using an ATmega328 microcontroller with a 5V operating Voltage, 16 MHz clock speed, 10-bit ADC, Baud rate of 4800, resulting in a 53 Hz sampling frequency. The conditions of measurement included:

- 1 minute baseline with no weight.
- 1 minute with 500 g weight added.
- 1 minute with additional 500 g weight added, total of 1,000 g.
- 1 minute with additional 500 g weight added, total of 1,500 g.
- 1,500 g weight removed, and 12,000 g weight added for the textile sensor conditions.
- 12,000 g weight removed, and 50 g weight added.

Measurements were recorded based on applying different weights over sixty second intervals. The graph demonstrates advantages of the sensor with the textile semi-conductive layer having signal consistency with minimized drift as compared to the FSR, a higher depth signal range of measuring 12 kg, measuring the same conditions even while the sensor is folded, and the ability to measure light 50g weight.

## VI. CONCLUSIONS AND FUTURE WORK

One of the final frontiers of industrial automation and robotics is human-like tactile sensing. There are continuously new technologies emerging [12]; however, many current approaches are at their sensing limits. This is demonstrated by the apparent emergence of vision-based technologies as the state-of-the-art for tactile sensing.

What has been demonstrated is a pioneering step forward in advancing tactile sensing. Humans continuously feel their environment, and the semi-conductive layer enables three key advantages. Active-Dynamic sensing, Multi-modal sensing (the ability to operate with two parameters of resistive and capacitive sensing), and dynamic form factors.

Through the introduction of high-depth signals, the need for integrating sensors and post-processing multiple data feeds is reduced. This novel approach provides a single low-cost sensor that is easy to manufacture, while reducing materials and parts, benefiting the environment with recyclability and lowering the carbon footprint. It allows machines and objects to interact with their environments with graceful, natural movements, by wielding human-like touch as primary feedback.

The future of the technology is promising. Further areas of research studies will be needed to fully understand its full potential. It presents a new ability in tactile sensing that will open up new industrial applications.

### ACKNOWLEDGMENT

E. Shim thanks Markus Van Kempen, Venture Capitalist in Residence, IBM Corporate Technical Strategy, for all his support and mentorship to reach this stage. E. Shim and C. Fraser thank Ludo Stellingwerff, of Co-existence and Almende Research, for his technical guidance and advice.

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
