# OpenReview forum: "Emerging High-Depth Signal Tactile Sensing Technology for Achieving Artificial Human touch"
_IEEE.org/IROS/2025/Workshop/Tactile_Sensing — IROS 2025 Workshop Tactile Sensing Poster_

### Official Review · Reviewer_nthE · 2025-09-16
**A novel tactile sensing technique**

**Rating:** 7
**Confidence:** 4

**Review:**

The authors discuss the limitations of current tactile sensors and propose a novel design capable of active-dynamic and multi-modal sensing, without being constrained by sensor deformation.

The reviewer suggests including figures of the sensor prototypes to improve clarity and help the audience better understand the proposed design.

---

### Official Review · Reviewer_btEw · 2025-09-25
**Major revisions needed for figures and paper length**

**Rating:** 6
**Confidence:** 4

**Review:**

This paper presents a novel tactile sensor design with a semi-conductive layer, claiming several advanced capabilities. However, a few critical revisions are required. The term “high-depth” is unclear and should be defined. The figures are a significant concern due to their poor quality and relevance; specifically, Figure 1 could be replaced with a diagram of the proposed sensor mechanism, and Figure 2 appears to be taken from a patent, which is inappropriate for a research paper. Figs 3 and 4 also need refinement. In addition, the manuscript must be shortened to meet the two-page limit (excluding references).

---

### Official Review · Reviewer_mFhA · 2025-09-25
**Adding more experimental results can enhance the contribution**

**Rating:** 7
**Confidence:** 5

**Review:**

This work have proposed one kind of tactile sensor that can provide a dynamic sensing range. This work is interesting. However, only few experimental results have been illustrated in the manuscript. It is better to provide more experimental details and results.